# COVID-19 Vaccination Is Not Associated with Psychiatric Adverse Events: A Meta-Analysis

**DOI:** 10.3390/vaccines11010194

**Published:** 2023-01-16

**Authors:** Sang-Eun Lee, Sung-Ryul Shim, Jung-Hae Youn, Hyun-Wook Han

**Affiliations:** 1Graduate School of Clinical Counseling Psychology, CHA University, Seongnam 13488, Republic of Korea; 2Department of Health and Medical Informatics, Kyungnam University, College of Health Sciences, Changwon 51767, Republic of Korea; 3Evidence Based Research Center, Kyungnam University, Changwon 51767, Republic of Korea; 4Department of Biomedical Informatics, CHA University School of Medicine, CHA University, Seongnam 13488, Republic of Korea; 5Institute of Basic Medical Sciences, School of Medicine, CHA University, Seongnam 13488, Republic of Korea; 6Institute for Biomedical Informatics, School of Medicine, CHA University, Seongnam 13488, Republic of Korea; 7Healthcare Big-Data Center, Bundang CHA Hospital, Seongnam 13488, Republic of Korea

**Keywords:** COVID-19, vaccination, adverse events, psychiatric disorders, depression, anxiety

## Abstract

Coronavirus disease 2019 (COVID-19) has been a global health problem since December 2019. Vaccination has been widely considered the best way to prevent COVID-19 pandemic, but public concerns about the safety of vaccines remain. There have been many studies reporting adverse events in the vaccinated. However, to date, no meta-analysis of the association of COVID-19 vaccination with psychiatric adverse events has been conducted yet. In this meta-analysis, studies on depression, anxiety and distress after COVID-19 vaccination were searched in the PubMed, Cochrane and Embase from January 2020 to April 2022. The OR of depression in four studies with a total sample size of 462,406 is obtained as 0.88 (95% CI; 0.75, 1.03), and the OR of anxiety as 0.86 (95% CI; 0.71, 1.05). However, there were no statistically significant differences between the groups. The mean difference of distress in two studies was −0.04 (95%CI; −0.05, −0.02; *p* < 0.0001). As a result of the moderator analysis, married people experienced less depression and anxiety after vaccination, and in White people, depression after vaccination was lower than others. We also found that people with a history of COVID-19 infection were more depressed and anxious after vaccination. We suggest that COVID-19 vaccination was not associated with a worsening of depression and anxiety.

## 1. Introduction 

Coronavirus disease 2019 (COVID-19) has been a global health problem since December 2019. Despite various efforts to solve the disease, they were insufficient in alleviating the physical and mental distress of patients. As of 1 November 2022, there have been 627,926,559 confirmed cases, including 6,573,145 deaths, while 4,972,809,715 people have been vaccinated, according to the World Health Organization (WHO) report [1].

In the last two years, global efforts against the COVID-19 pandemic have accelerated the approval of vaccines and their dissemination for vaccination [2,3]. However, public concerns about the effect or safety of vaccination still exist, such as the effectiveness of vaccines, types and severity of adverse reactions after vaccination, age of vaccine recipients and number of doses [4]. It is noted that adverse events after COVID-19 vaccination include diverse physical symptoms as well as psychiatric illness [5,6,7]. Fatal adverse events include myocarditis, pericarditis and blood clots, and other symptoms continue to be reported, including pain at the injection site, fever, nausea, fatigue, headache, depression, anxiety, irritability and insomnia [6,8,9].

Influenza, a respiratory disease like COVID-19, may be regarded as a trigger for psychosis [10]. This approach of viewing mental symptoms in terms of neuropsychiatric sequelae could help predict immune responses in COVID-19 vaccine recipients [6]. It is important to monitor psychotic adverse events in patients because the potential psychiatric sequelae of vaccines may be exacerbated with environmental factors, e.g., disasters that cause psychological vulnerability. 

In particular, depression and anxiety are the most common symptoms among psychiatric disorders. Clinical characteristics of the two symptoms may appear simultaneously in patients [11]. Therefore, it is important to track the changes in depression and anxiety after vaccination.

Moreover, as the prolonged COVID-19 pandemic has worsened the mental health of the public [12,13,14], the correlation between vaccination and mental health is of particular interest. If the relationship between COVID-19 vaccination and risks on mental illness becomes more predictable, we will be able to reduce the psychological burden of vaccination under the current COVID-19 pandemic situation.

In the present meta-analysis based on studies conducted in the early stages of the pandemic, we derived an integrated result revealing changes in mental disorders after COVID-19 vaccination. Our evidence-based results will contribute to alleviating public concerns surrounding COVID-19 vaccines and the development of COVID-19 policy.

## 2. Materials and Methods

This meta-analysis was registered at PROSPERO (CRD42023390999) and guided by the standard Preferred Reporting Items for Systematic Reviews and Meta-Analyses (PRISMA) protocol [15] (Appendix A). 

### 2.1. Searching Strategy

The study search was conducted from database inception to 11 April 2022. Our studies were identified from a search in PubMed and Cochrane Library for articles that were published from 1 January 2020 through to 11 April 2022. The search initially used the following strings: Search (“COVID-19 Vaccines”[Mesh] OR “COVID-19 Vaccines”[tiab] OR “mRNA Vaccine” [Supplementary Concept]) AND (“Mental Disorders”[Mesh] OR “Depression”[Mesh] OR “Depressive Disorder”[Mesh] OR “Depression”[tiab]) AND (2020/01/01:3000/12/12[pdat]). 

Screening criteria were comparison (vaccinated or unvaccinated), outcome measurement (e.g., PHQ-2, PHQ-4, PHQ-8, PHQ-9, GAD-2, GAD-7, PROMIS-29) and article type (prospective cohort study, panel study, cross-sectional study). Language restrictions were not imposed.

### 2.2. Study Selection

Inclusion criteria were established prior to article reviews and were as follows [16]: (1) participants confirmed to have been COVID-19-vaccinated or non-vaccinated; (2) exposed cohort/controls consisted of COVID-19-vaccinated and non-vaccinated participants; (3) mental disorder as outcome using standardized measures (e.g., PHQ-2, PHQ-4, PHQ-8, PHQ-9, GAD-2, GAD-7, PROMIS-29); (4) human-based studies. 

Exclusion criteria were as follows: (1) no use of standardized measures for mental dis order symptoms as outcomes of interest; (2) case study; (3) non-human studies.

### 2.3. Types of Exposures and Outcomes

The exposed cohorts received one dose or two doses of COVID-19 vaccination and the control cohorts did not receive any dose of vaccination. The outcomes in this study were symptoms of depression, anxiety and distress assessed by self-report clinical scales. 

### 2.4. Measures

The scale Patient Health Questionnaire (PHQ) is used to diagnose depression symptoms [17]. The studies using PHQ adopted different versions of the questionnaire according to the number of items: PHQ-2, PHQ-8 and PHQ-9 [18,19,20,21]. PHQ-4 is a brief screening tool for depression and anxiety [22]. This scale was used in two studies for rating distress symptoms [23,24]. General Anxiety Disorder (GAD) is a measurement tool for anxiety symptoms [25]. This scale was also used in studies in two versions according to the number of items included: GAD-2 and GAD-7 [18,19,20,21]. Patient-Reported Outcomes Measurement Information System (PROMIS-29) is a scale for rating physical health and mental health factors [26]. It contains seven domains, including depressive symptoms and anxiety [21]. 

### 2.5. Types of Moderators

A variety of parameters were used in this analysis. We conducted moderator analyses by number of participants, follow up period, age, sex, race, measures, marital status and history of COVID-19. We searched for how differences in some of these moderators affect outcomes.

### 2.6. Quality Assessment

The Newcastle–Ottawa Scale (NOS) was used to assess the quality of cohort studies included in this meta-analysis [27]. We evaluated the following 3 domains: (1) Selection, (2) Comparability and (3) Outcome. The first domain contains 4 items to confirm the representativeness of the exposed cohort and the non-exposed cohort. The second domain, Comparability, contains one item related to analysis for the comparison of cohorts. The adequacy of outcome and follow up is assessed in the Outcome domain, with 3 items. The item of Comparability is allowed up to two stars and the others are awarded a maximum of one star each. The sum of the total scores was rated as good, fair or poor. 

### 2.7. Statistical Analysis

We used Odds Ratio (OR) for the analysis of mental disorder symptoms, i.e., depression and anxiety, due to different standardized measures used in studies. Mean difference was used for analysis associated with distress symptoms. ORs were calculated using random-effects models and MDs were calculated using a fixed-effect models as in model assumptions with 95% CI. Moreover, we analyzed subgroups regarding dosage of vaccine: one dose, two doses, and one or two doses using random effects models. Meta-analysis was performed with a statistical software, R. Statistical significance was set at *p* < 0.05.

### 2.8. Assessment of Heterogeneity

We assessed statistical heterogeneity with Cochran’s Q test and the I^2^ statistic. In case of statistically significant heterogeneity (*p* < 0.1 in Cochran’s Q or I^2^ > 50%), we used the random-effects model to calculate the OR with 95% CI.

### 2.9. Assessment of Publication Bias

Potential publication bias was assessed by a funnel plot of this meta-analysis using standard error as the measure of study size and odds ratio of measures. We used Egger’s test (linear regression test of funnel plot asymmetry) and Begg’s test (rank correlation test of funnel plot asymmetry) to evaluate potential publication bias.

## 3. Results

### 3.1. Study Selection

The initial search yielded 731 potentially relevant articles (PubMed: *n* = 112, Embase: *n* = 619, Cochrane: *n* = 0). Removing 43 duplicates, 26 studies were eligible after screening titles and abstracts. After intensive screening, nine studies were found to be eligible. Finally, six studies that met all inclusion criteria were included [18,19,20,21,23,24]. The six studies consisted of 478,523 subjects: 246,574 in exposed cohorts, 223,859 in non-exposed cohorts [18,19,20,21,23] and one study contained a total of 8090 subjects [24]. A flow chart below shows the study identification and process of study selection (Figure 1).

### 3.2. Characteristics of the Included Studies

Table 1 shows the characteristics of the included studies. Four studies were conducted in the USA [19,21,23,24], and the remaining two studies were conducted in Peru [18] and Sweden [20]. One study was a cross-sectional study [19] and four of those were prospective cohort studies [18,20,21,24]. The other was a panel study [23].

### 3.3. Quality Assessment

Table 2 shows the NOS of the included studies. The six studies received a total score of 7 or 8, which indicates good quality of the study [18,19,20,21,23,24]. In these studies, the representativeness of the cohort (vaccinated and non-vaccinated) was adequately supported, and analysis of variables was also performed appropriately.

We were unable to ascertain detailed information from some studies, such as dropout rates [19,21,23,24]; however, they were sufficient to obtain scores for good quality in outcome and follow-up.

### 3.4. Outcome Findings

#### 3.4.1. Depression

Four studies (*n* = 462,406; 244,931 exposed group and 217,475 control group) reported changes in depression estimated by PHQ-2, PHQ-8, PHQ-9 and PROMIS-29 [18,19,20,21]. The OR of depression for the ever-vaccinated group versus the never-vaccinated group was 0.88 (95% CI; 0.75, 1.03). There were no statistically significant differences between groups. Two studies used PHQ-2 to assess for depression [18,19] and another used PHQ-9 as a scale [20]. The other study used PHQ-8 and PROMIS-29 v2.0 Scale [21]. In subgroup analyses, we found that the vaccinated compared to the non-vaccinated had low depressive symptoms (OR, 0.87 [95%CI; 0.71, 1.07] in one dose; OR, 0.94 [95%CI; 0.64, 1.38] in two doses; OR, 0.83 [95%CI; 0.80, 0.86] in one or two doses) but without statistical significance (Figure 2).

#### 3.4.2. Anxiety

Four studies (*n* = 462,406; 244,931 exposed group and 217,475 control group) reported changes in anxiety estimated by GAD-2, GAD-7 and PROMIS-29 [18,19,20,21]. The OR of anxiety for the ever-vaccinated group versus the never-vaccinated group was 0.86 (95% CI; 0.71, 1.05). There were no statistically significant differences between groups. Two studies used GAD-2 to assess for anxiety [18,19] and other two studies used GAD-7 as scale [20,21]. One study used PROMIS-29 v2.0 Scale [21]. In subgroup analyses, the vaccinated compared to the unvaccinated had less anxiety symptoms (OR, 0.95 [95%CI; 0.67, 1.34] in one dose; OR, 0.79 [95%CI; 0.54, 1.35] in two doses; OR, 0.85 [95%CI; 0.83, 0.88] in one or two doses). It was not statistically significant (Figure 3).

#### 3.4.3. Distress

Two studies(*n* = 16,117) reported changes in mental distress with PHQ-4, rated with two items for depressive symptoms and two items for anxiety symptoms [23,24]. The mean difference of distress for the vaccinated group, when compared to the unvaccinated group, was −0.04 (95%CI; −0.05, −0.02; *p* < 0.0001), indicating that vaccination against COVID-19 was effective in alleviating distress symptoms (Figure 4).

### 3.5. Moderator Analysis

We considered potential moderating roles of the following variables using meta-regression and meta-ANOVA models: number of participants; follow up period; age; sex; race; measures; marital status; COVID-19 history. Table 3 provides an overview of the moderator analyses of depression and anxiety.

#### 3.5.1. Depression

The result of analysis in depression symptoms revealed significance in the following variables: follow up period, race—white, measures, marital status and COVID-19 history. Depression symptoms increased significantly as the follow up period of the study increased. Studies with more than 70% White people showed fewer depressive symptoms among the vaccinated compared to those not vaccinated than other studies (OR, 0.78 [95%CI; 0.69, 0.88; *p* = 0.008]). With respect to a scale for assessing depression symptoms, in studies using PHQ-2 and PHQ-9, the vaccinated showed lower depressive symptoms compared to the non-vaccinated (PHQ-2; OR, 0.83 [95%CI; 0.78, 0.88], PHQ-9; OR, 0.71 [95%CI; 0.60, 0.83], *p* = 0.015). In contrast, studies using PHQ-8 and PROMIS-29 showed a worsening of depression symptoms in the vaccinated compared to the unvaccinated (PHQ-8; OR, 1.10 [95%CI; 0.84, 1.43], PROMIS-29; OR, 1.02 [95%CI; 0.79, 1.33], *p* = 0.015). We found moderating effects on marital status, where studies with more than 60% married people found lower depressive symptoms in the vaccinated compared to the unvaccinated (OR, 0.71 [95%CI; 0.56, 0.89; *p* = 0.038]). Additionally, studies with more than 50% of participants with a history of COVID-19 infection were found to be higher in depression symptoms in the vaccinated compared to the unvaccinated (OR, 1.06 [95%CI; 0.87, 1.29; *p* = 0.011]).

#### 3.5.2. Anxiety

The result of analysis in anxiety symptoms also revealed significance in the following variables: follow up period, sex—female, marital status and COVID-19 history. Studies have shown an association between the length of the follow-up period and an increase in anxiety symptoms. Studies with more than 70% female participants showed lower anxiety symptoms among the vaccinated compared to those not vaccinated than other studies (OR, 0.65 [95%CI; 0.50, 0.83; *p* = 0.006]). With respect to marital status, studies with more than 60% married people found lower anxiety symptoms in the vaccinated compared to the unvaccinated (OR, 0.64 [95%CI; 0.48, 0.85; *p* = 0.017]). Additionally, studies with more than 50% of participants with a history of COVID-19 infection were found to be higher in anxiety symptoms in the vaccinated compared to the unvaccinated (OR, 1.11 [95%CI; 0.88, 1.40; *p* = 0.007]).

### 3.6. Assessment of Heterogeneity

There was moderate heterogeneity between studies in depression analysis (I^2^ = 45.5%, *p* = 0.066 in Cochran’s Q) and there was substantial heterogeneity between studies in anxiety analysis (I^2^ = 61.9%, *p* = 0.072 in Cochran’s Q). Therefore, we used a random effects model to calculate the ORs and conducted research focusing on the analysis of diverse variables that explain heterogeneity.

### 3.7. Publication Bias

The Funnel plot to detect the publication bias in the included studies is summarized in Figure 5. In the depression analyses, two studies lay to the left of the funnel. Individual studies are distributed toward the top of the graph and two of them are at the bottom. However, there was no evidence of publication bias in this meta-analysis (Egger’s test, *p* = 0.33, Begg’s test, *p* = 0.14). In the anxiety analyses, one study lay to the left and one study lay to the right of the funnel. There was also no evidence of publication bias in this analysis (Egger’s test, *p* = 0.94, Begg’s test, *p* = 0.53).

## 4. Discussion

To our knowledge, this is the first meta-analysis study to assess psychiatric adverse events after vaccination against COVID-19 using early stages data of the pandemic. We hypothesized that COVID-19 vaccination may link to changes in psychiatric symptoms. Here, we found that distress, a concept that includes symptoms of depression and anxiety, was associated with 4% lower probability in people who have received the COVID-19 vaccine. However, no significant association was found between COVID-19 vaccination and depression as well as anxiety. Among early four works in this study, the two studies reported positive change of depression and anxiety symptoms after receiving COVID-19 vaccine in adults [19,20], while the other two studies reported no significant change of depression or anxiety symptoms after vaccination [18,21]. Similarly, among other studies on the link between COVID-19 vaccines and mental disorders, some reported the results that vaccines have a positive effect on mental health [28,29]. They observed changes in psychological symptoms, such as anxiety, depression, psychological distress as well as displeasure after vaccination, and reported that there were rare psychological side effects on vaccination. They also suggested some potential factors of vaccination that improve the mental health of recipients, for example, a sense of security from severe illness and death due to infection and the expectation of a return to daily life from social isolation, etc. Some studies published after the search period of this research also supported the view that COVID-19 vaccines do not exacerbate psychiatric symptoms [30,31,32]. 

In this meta-analysis, we concluded that COVID-19 vaccination does not appear to affect depression and anxiety. It supports the importance of COVID-19 vaccination in terms of immunological prevention of COVID-19 infection. In addition, we suggest that it contributes to relieving public concerns and to some extent answering their questions about psychiatric adverse events in relation to vaccine safety and hesitation in vaccination. 

Interestingly, this study has several important findings in moderator analyses. First, we found that married people were less depressed and anxious after COVID-19 vaccination. This result is consistent with previous studies that concluded married groups were more likely to be healthier with regard to psychiatric disorders. Some longitudinal studies showed that marriage is associated with declines in psychological distress and increase in psychological wellbeing [33,34]. Further, major depressive disorder (MDD) was found to be correlated with separated/divorced marital status [35]. Moreover, married patients diagnosed with depression had better prognoses than those not married [36]. This suggests that marriage is related to providing personal social integration and emotional support that enhances mental health, especially during disasters and health crises, such as the current pandemic situation.

Second, it was found that people with a history of COVID-19 infection had more severe symptoms of depression and anxiety after being vaccinated. This result is in accord with the results of previous studies that suggest there is a higher prevalence of depression or anxiety in patients with COVID-19 [37,38,39,40]. The causes of these psychiatric disorders after infection are likely to be based on multiple factors. One of the factors is limiting and severing relationships with close people, such as family, loved ones, and friends. Other factors include the uncertainty about the ongoing pandemic, the fear caused by daily reports of COVID-19 infection and related deaths, and the confusion caused by inaccurate information. 

Third, we found that White people had fewer depressive symptoms after COVID-19 vaccination compared to other races. This result is also to some extent consistent with previous studies that conclude that White people are less depressive than Black, Hispanic, and other people during the COVID-19 pandemic [41,42,43]. These racial differences are influenced by social vulnerabilities, such as low level of education and income, and lack of adequate health care [44]. Ethnic inequality is not an issue that has just emerged, however disaster situations become a factor that further expose the gap of inequality [45]. The distress of people experiencing racism and stigma due to COVID-19 negatively affects their mental health [46].

This study includes limitations that other meta-analyses have in common. The results of the meta-analysis could not be based on direct evidence and are affected by the bias of the original studies. The number of six studies included in the meta-analysis is limited enough to draw valid results. This requires an additional search of studies published from April 2022 to the present. As another limitation, it is also pointed out that the symptoms to be investigated may be a mixture of psychiatric symptoms and non-psychiatric symptoms. The six studies measured both mental symptoms and vaccination status relying on self-report, indicating the lack of clinical confirmation. They also used different scales to identify the same disease. However, the scales are composed of similar or identical items for describing the symptoms. The evaluation tools are practical scales that aid in psychiatric diagnosis in the clinical settings. Despite these limitations, the result of this study provides important information on the safety of COVID-19 vaccination on psychiatric symptoms.

In the ongoing COVID-19 era, studies investigating the impact of vaccines are still actively underway. Therefore, additional studies of larger population are required to evaluate the relationship between COVID-19 vaccination and psychiatric problems.

## 5. Conclusions

According to our analysis, it can be concluded that COVID-19 vaccination is not associated with depression and anxiety as psychiatric adverse events. Rather, the people vaccinated against COVID-19 experienced a 4% reduction in distress. After vaccination, married people were less depressed and anxious, and those with a history of COVID-19 infection were more depressed and anxious. In the current pandemic, vaccination against COVID-19 is a safe precaution for the physical and mental health of the public.

## Figures and Tables

**Figure 1 vaccines-11-00194-f001:**
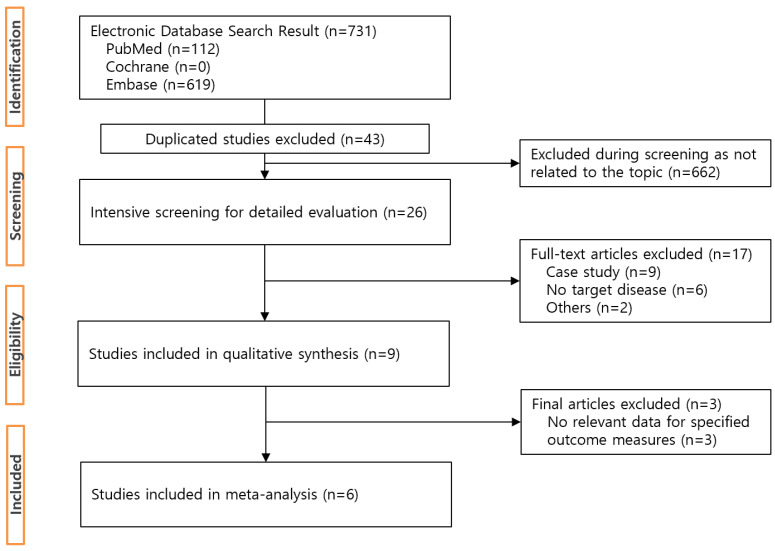
Flowchart of study selection process.

**Figure 2 vaccines-11-00194-f002:**
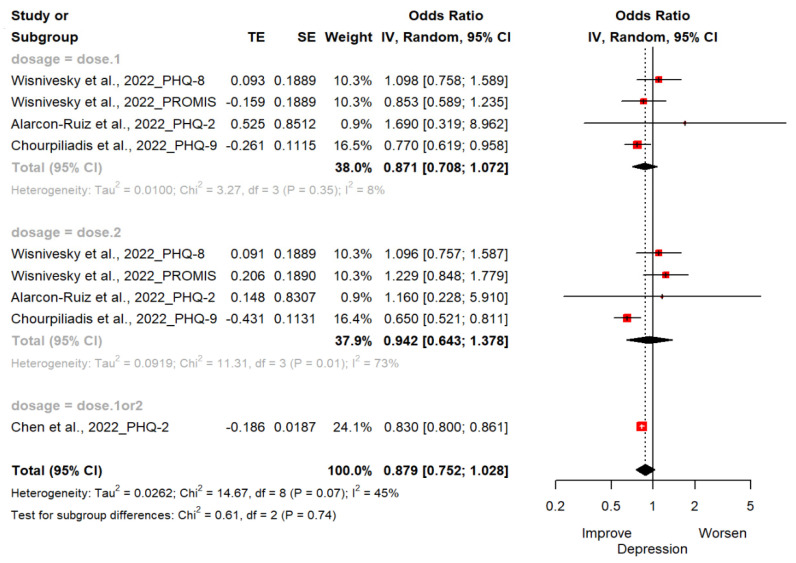
Forest plot diagram showing the effect of COVID-19 vaccination on depression. Odds ratio; CI, confidence interval [18,19,20,21].

**Figure 3 vaccines-11-00194-f003:**
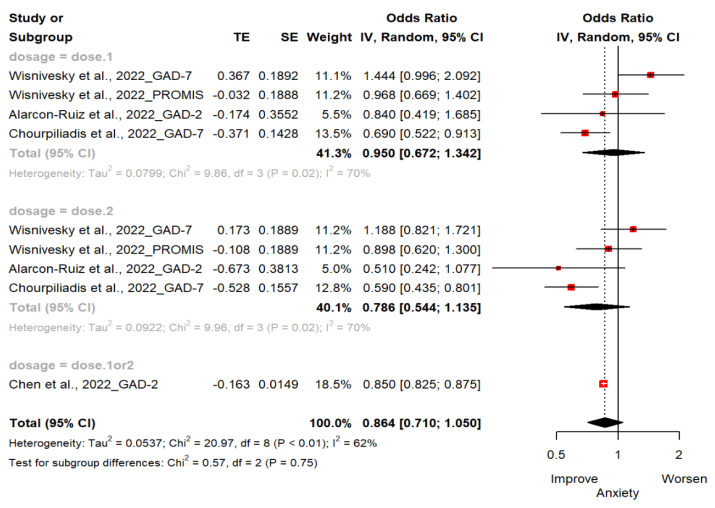
Forest plot diagram showing the effect of COVID-19 vaccination on anxiety. Odds ratio; CI, confidence interval [18,19,20,21].

**Figure 4 vaccines-11-00194-f004:**
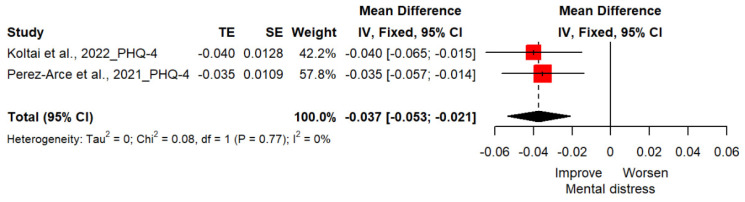
Forest plot diagram showing the effect of COVID-19 vaccination on distress. Mean difference; CI, confidence interval [23,24].

**Figure 5 vaccines-11-00194-f005:**
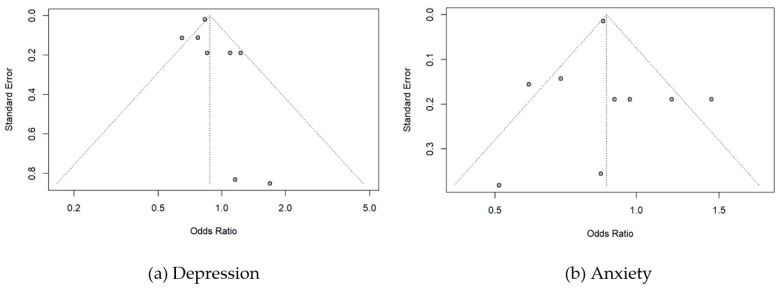
Funnel plots for publication bias of odds ratio on (**a**) Depression and (**b**)Anxiety.

**Table 1 vaccines-11-00194-t001:** Characteristics of included studies.

Study	Country	Study Design	Sample Size (Mean Age)	Population (Exposed Cohort, Non-Exposed Cohort)	Follow Up Period	Exposure		Outcome Measurements		
						Exposed Cohort	Non-Exposed Cohort	Depression	Anxiety	Distress
Wisnivesky et al., 2022 [21]	USA	prospective cohort	453 (49.9)	COVID-19 patients (324, 129)	6 months	COVID-19 vaccinated	COVID-19 unvaccinated	PHQ-8, PROMIS-29	GAD-7, PROMIS-29	–
Alarcon-Ruiz et al., 2022 [18]	Peru	prospective cohort	861 (72.2)	older adults in Peru (794, 67)	1 month	COVID-19 vaccinated	COVID-19 unvaccinated	PHQ-2	GAD-2	–
Chen et al., 2022 [19]	USA	Cross -sectional	453,167 (55.0)	adults in USA (236,757, 216,410)	-	COVID-19 vaccinated	COVID-19 unvaccinated	PHQ-2	GAD-2	–
Chourpiliadis et al., 2022 [20]	Sweden	prospective cohort	7925 (53.4)	adults in Sweden (7056, 869)	1 month	COVID-19 vaccinated	COVID-19 unvaccinated	PHQ-9	GAD-7	–
Koltai et al., 2022 [24]	USA	prospective cohort	8090 (51.0)	adults in USA (–)	2 weeks	COVID-19 vaccinated	COVID-19 unvaccinated	–	–	PHQ-4
Perez-Arce et al., 2021 [23]	USA	panel	8027 (53.8)	adults in USA (1643, 6384)	2 weeks	COVID-19 vaccinated	COVID-19 unvaccinated	–	–	PHQ-4

GAD-2: General Anxiety Disorder 2-Item Scale, GAD-7: General Anxiety Disorder 7-Item Scale, PHQ-2: Patient Health Questionnaire Depression Module-2, PHQ-4: Patient Health Questionnaire Depression Module-4, PHQ-8: Patient Health Questionnaire Depression Module-8, PHQ-9: Patient Health Questionnaire Depression Module-9, PROMIS-29: Patient-Reported Outcomes Measurement Information System-29 v2.0 Scale. Note: Cells containing “–” indicate that the study author did not provide any relevant information for that column.

**Table 2 vaccines-11-00194-t002:** Newcastle-Ottawa quality assessment.

□	□	Selection			□	Comparability		Outcome □				Total Score	Quality Power
□	[A]	[B]	[C]	[D]	□	[E]	□	[F]	[G]	[H]	□		
Wisnivesky et al., 2022 [21]	★	★	★	★	□	★★	□	★	★	□	□	★★★★★★★★ (8)	Good
Alarcon-Ruiz et al., 2022 [18]	★	★	★		□	★★	□	★	★	★	□	★★★★★★★★ (8)	Good
Chen et al., 2022 [19]	★	★	★		□	★★	□	★	★	□	□	★★★★★★★ (7)	Good
Chourpiliadis et al., 2022 [20]	★	★	★		□	★★	□	★	★	★	□	★★★★★★★★ (8)	Good
Koltai et al., 2022 [24]	★	★	★	★	□	★★	□	★	★	□	□	★★★★★★★★ (8)	Good
Perez-Arce et al., 2021 [23]	★	★	★	★	□	★★	□	★	★	□	□	★★★★★★★★ (8)	Good

[A] Representativeness of the exposed cohort, [B] selection of the non-exposed cohort, [C] ascertainment of exposure, [D] outcome of interest not present at start of study, [E] comparability of cohorts on the basis of the design or analysis, [F] ascertainment of outcome, [G] adequacy of duration of follow-up, [H] adequacy of completeness of follow-up. A study can be awarded a maximum of one star for each numbered item for Selection/Outcome and a maximum of two stars can be awarded for Comparability. Good quality: 3 or 4 stars in selection domain AND 1 or 2 stars in comparability domain AND 2 or 3 stars in outcome domain. Fair quality: 2 stars in selection domain AND 1 or 2 stars in comparability domain AND 2 or 3 stars in outcome domain. Poor quality: 0 or 1 star in selection domain OR 0 stars in comparability domain OR 0 or 1 stars in outcome domain.

**Table 3 vaccines-11-00194-t003:** Associations of moderators of depression and anxiety.

Depression						Anxiety					
Variables	*k*	Coefficient (95% CI)	OR (95% CI)	*p*-Value *	*p*-Value †		*k*	Coefficient (95% CI)	OR (95% CI)	*p*-Value *	*p*-Value †
No. of participants	9	−0.00 (−0.00 to 0.00)		0.679			9	−0.00 (−0.00 to 0.00)		0.932	
Follow up period	8	0.02 (0.01 to 0.03)		0.001			8	0.03 (0.01 to 0.04)		<0.001	
Age					0.441						0.366
>60	2		1.39 (0.43 to 4.56)				2		0.66 (0.36 to 1.22)		
<60	7		0.87 (0.75 to 1.02)				7		0.89 (0.72 to 1.10)		
Female rate	9				0.060		9				0.006
>70%	4	−1.20 (−2.78 to 0.37)	0.73 (0.58 to 0.91)	0.134			4	−2.04 (−3.93 to −0.16)	0.65 (0.50 to 0.83)	0.034	
Others	5		0.95 (0.81 to 1.12)				5		0.99 (0.83 to 1.19)		
Race					0.008						0.497
White	3	−0.92 (−1.48 to −0.35)	0.78 (0.69 to 0.88)	0.001			3	0.36 (−0.17 to 0.88)	0.77 (0.52 to 1.14)	0.182	
Others	6		1.07 (0.88 to 1.30)				6		0.91 (0.70 to 1.16)		
Married					0.038						0.017
Yes	2		0.71 (0.56 to 0.89)				2		0.64 (0.48 to 0.85)		
No	7		0.96 (0.81 to 1.13)				7		0.96 (0.81 to 1.13)		
History of COVID-19					0.011						0.007
Yes	4		1.06 (0.87 to 1.29)				4		1.11 (0.88 to 1.39)		
No	5		0.78 (0.69 to 0.88)				5		0.73 (0.61 to 0.88)		
Measures					0.015						0.791
PHQ-2	3		0.83 (0.78 to 0.88)			GAD-2	3		0.76 (0.50 to 1.17)		
PHQ-8	2		1.10 (0.84 to 1.43)			GAD-7	4		0.90 (0.64 to 1.25)		
PHQ-9	2		0.71 (0.60 to 0.83)			PROMIS-29	2		0.93 (0.58 to 1.51)		
PROMIS-29	2	□	1.02 (0.79 to 1.33)	□	□	□	□	□	□	□	□

k, number of observations; OR, odds ratio; coefficient, regression coefficient. * *p*-values from meta-regression analysis using the restricted maximum likelihood. † *p*-values from meta-ANOVA for categorical moderators.

## Data Availability

Data are contained within the article or Appendix A.

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
