# Peer review of "COVID-19 Vaccination Is Not Associated with Psychiatric Adverse Events: A Meta-Analysis"

_vaccines, 2023, doi:10.3390/vaccines11010194_

Round 1

Reviewer 1 Report

Thank you for the invitation to review this manuscript. I appreciate the efforts of the authors. The authors have conducted a meta-analysis on 6 studies ascertaining the impact of COVID-19 on psychological health. However, the claim of authors is not robust, and cannot provide a firm conclusion that the administration of COVID-19 vaccines can alleviate depression, anxiety and distress among recipients. There are several other co-variates that may relate to psychological disturbances i.e., previous COVID-19 infections, socio-economic status, fear of getting infection etc. The time between data collection and vaccination receiving date also plays an important role in these findings. The number of studies is very few, mostly from the USA, and this is a major concern for sub-group anlaysis. The findings cannot be generalized from a global perspective. Moderator analyses showed various factors associated with psychological illnesses among vaccinees, but the author did not discuss the plausible association of these factors with the development of anxiety, depression and stress. The follow-up data is also missing in primary studies used for meta-analyses. These are some serious limitations of this review which should be considered during the analyses, interpretations of findings and drafting the discussion section.

The authors are stating the incidence of psychological illness among vaccine recipients throughout the manuscript and claimed that vaccines do not cause any such problem in the conclusion section. The rational of the study is very weak. What prompted the authors to conduct this analysis, how the review question was generated? Can the authors explain in detail that how covid-19 vaccines can improve the psychological health? Is it due to perceived safety against covid-19 after getting infection? A possible mechanism should be discussed in the discussion section.

I believe that the title is not much catchy and tags the vaccines with such side effects. I suggest the authors to change it as

COVID-19 Vaccination is not associated with Psychiatric Adverse Events: A Systematic Review and Meta-Analysis

Reviewer 2 Report

Dear Authors, please take into your consideration the comments above.

Major revisions

Since the literature search was completed until April 11, 2022, it is probable that further studies have been published on this topic. With a quick search, I find the following studies:

DOI: 10.1177/07334648221092029

DOI: 10.3389/fpubh.2022.819062

Thus, I suggest updating your review until now in order to get more valid results.

Please, expand the Introduction section in order to make readers familiar with your topic. For example, you could describe some psychiatric events such as depression and anxiety, especially during the pandemic.

Minor revisions

Please, add the results for the distress in the abstract.

In the section “3.1. Study Selection”, please present the text before the flowchart.

As you mention, you used Egger’s test (linear regression test of funnel plot asymmetry) and Begg’s test (rank correlation test of funnel plot asymmetry) to evaluate potential publication bias. Please present these results in the section “3.7. Publication bias”.

There is confusion regarding the references. For example, in Table 1 the first author for the study in Peru is Christoper but in the references the first name is Alarcon-Ruiz. Please, check all the references.

In the Discussion section, you write “To our knowledge, this is the first meta-analysis study to assess psychiatric adverse events after vaccination against COVID-19 using national representative data”. I think that this is wrong since studies with 453 (Juan P et al., 2022) and 861 participants (Christoper A et al., 2022) could not be conducted with national representative data. The same with the sentence “Studies included in this meta-analysis consisted of large-scale population surveys”.

Please, add the limited number of studies (n=6) as a limitation.

A native English speaker could improve the English language and style of your manuscript. There are errors or misunderstandings in several sentences. For example, please see the following sentences:

PubMed and Cochrane Library search for studies from January 01, 2020 to April 11, 2022 were selected using electronic database search formula.

The studies included in this analysis have differed in a number of parameters.

This meta-analysis composed of cohort studies was rated as the Newcastle-Ottawa Scale (NOS) for quality assessment.

We sum up the total score and rate it as good, fair or pool.

The initial search identified a total of 731 articles from the electronic database

There was no statistically significant between groups.

Round 2

Reviewer 2 Report

Dear Authors thank you for the revised manuscript.